# Tailored Multiplex Real-Time RT-PCR with Species-Specific Internal Positive Controls for Detecting SARS-CoV-2 in Canine and Feline Clinical Samples

**DOI:** 10.3390/ani13040602

**Published:** 2023-02-09

**Authors:** Gyu-Tae Jeon, Hye-Ryung Kim, Jong-Min Kim, Ji-Su Baek, Yeun-Kyung Shin, Oh-Kyu Kwon, Hae-Eun Kang, Ho-Seong Cho, Doo-Sung Cheon, Choi-Kyu Park

**Affiliations:** 1Animal Disease Intervention Center, College of Veterinary Medicine, Kyungpook National University, Daegu 41566, Republic of Korea; 2Foreign Animal Disease Division, Animal and Plant Quarantine Agency, Gimcheon 39660, Republic of Korea; 3Bio-Safety Research Institute, College of Veterinary Medicine, Jeonbuk National University, Iksan 54596, Republic of Korea; 4Postbio Inc., Guri-si 11906, Republic of Korea

**Keywords:** SARS-CoV-2, multiplex real-time RT-PCR, *RdRp* gene, *N* gene, internal positive control

## Abstract

**Simple Summary:**

Given that SARS-CoV-2 infections in companion dogs and cats have been frequently reported worldwide during the ongoing COVID-19 pandemic, a multiplex real-time RT-PCR assay is urgently required to reliably detect SARS-CoV-2 infection in companion animals. In this study, we developed a tailored multiplex real-time RT-PCR assay to simultaneously detect *RdRp* and *N* genes of currently circulating SARS-CoV-2 variants and canine or feline *16S rRNA* as an endogenous internal positive control. The developed assay had high sensitivity, specificity, and accuracy and could detect all tested SARS-CoV-2 variants, including Omicron subvariants. Clinical evaluation of canine and feline specimens revealed that the diagnostic sensitivity of the assay was equivalent to that of a commercial SARS-CoV-2 multiplex real-time RT-PCR kit. Furthermore, canine or feline endogenous internal positive control was amplified using the developed assay while avoiding false-negative results. Considering the high sensitivity, specificity, accuracy, and reliability, the developed assay can help diagnose COVID-19 in dogs and cats and potentially play a vital role in the rapid diagnosis and control of SARS-CoV-2 infections in companion animals.

**Abstract:**

Severe acute respiratory syndrome coronavirus 2 (SARS-CoV-2) infections have been frequently reported in companion dogs and cats worldwide during the ongoing coronavirus disease. However, RT-qPCR methods developed for humans have been used for the diagnosis of SARS-CoV-2 infections in suspected companion dogs and cats owing to the lack of the companion animal-tailored methods. Therefore, we developed a multiplex RT-qPCR (mRT-qPCR) using newly designed primers and probes targeting *RdRp* and *N* genes of all currently circulating SARS-CoV-2 variants as well as the canine or feline *16S rRNA* gene as an endogenous internal positive control (EIPC) for reliable diagnosis of SARS-CoV-2 infection from suspected dogs and cats. The developed mRT-qPCR assay specifically detected the target genes of SARS-CoV-2 but no other canine or feline pathogens. Furthermore, canine and feline EIPCs were stably amplified by mRT-qPCR in samples containing canine- or feline-origin cellular materials. This assay has high repeatability and reproducibility, with an optimal limit of detection (<10 RNA copies per reaction) and coefficients of variation (<1.0%). The detection rate of SARS-CoV-2 of the developed mRT-qPCR was 6.6% for canine and feline nasopharyngeal samples, which was consistent with that of a commercial mRT-qPCR kit for humans. Collectively, the newly developed mRT-qPCR with canine and feline EIPC can efficiently diagnose and evaluate the viral load in field specimens and will be a valuable tool for etiological diagnosis, epidemiological study, and controlling SARS-CoV-2 infections in canine and feline populations.

## 1. Introduction

Severe acute respiratory syndrome coronavirus 2 (SARS-CoV-2), the causative agent of the ongoing coronavirus disease-2019 (COVID-19) pandemic, is a single-stranded positive-sense RNA virus that belongs to the genus *Betacoronavirus* in the subfamily *Orthocoronavirinae* of the family *Coronaviridae* [1] (for abbreviations, see Appendix A). Although it is considered a zoonotic virus that might have been derived from bats, it is now also considered a reverse zoonotic virus that can be transmitted from humans to various domestic and wild animals under natural and experimental conditions [2,3,4]. Notably, pet dogs and cats in close contact with their owners are highly vulnerable to the virus infection if their owners are infected [5,6]. Thus far, numerous cases of SARS-CoV-2 infection in dogs and cats have been reported worldwide, including in the Republic of Korea [7,8,9]. Although the role of pet animals in the spread of COVID-19 has not been confirmed, it raises concerns that the emergence of unexpected variants through host adaptation to these infected pet animals may pose a threat to public health [10,11,12].

Rapid and accurate diagnosis of SARS-CoV-2 infection is a prerequisite for disease control in humans and animals. Currently, reverse transcription real-time quantitative polymerase chain reaction (RT-qPCR) is accepted as the global standard method for detecting SARS-CoV-2 infection. RT-qPCR is based on primers and probes that specifically amplify targeted regions of conserved viral gene sequences, such as open reading frame 1ab (*ORF1ab*), RNA-dependent RNA polymerase (*RdRp*), nucleocapsid (*N*), envelope (*E*), or even the spike protein (*S*) gene [13]. To date, numerous RT-qPCR protocols have been developed and are widely used for detecting SARS-CoV-2 infection in the human population [14]. However, to the best of our knowledge, an RT-qPCR protocol for the diagnosis of SARS-CoV-2 infections in pet dogs and cats has not yet been reported, and RT-qPCR protocols developed for humans have been used in most animal disease diagnostic laboratories to diagnose SARS-CoV-2 infections in pet dogs and cats [4,5,6,8]. Notably, RT-qPCR methods for humans utilize human housekeeping genes, such as human *RNase P* or *glyceraldehyde 3-phosphate dehydrogenase* (*GAPDH*), as endogenous internal positive controls (EIPC) to monitor potential problems throughout the RT-qPCR process, such as sample collection, nucleic acid extraction, and subsequent test results [14,15]. However, the reliability of human-specific RT-qPCR methods for animal specimens cannot be guaranteed, as human EIPC cannot be amplified from animal specimens. Therefore, for a RT-qPCR method that screens SARS-CoV-2 infections in canine and feline clinical samples, a housekeeping gene stably expressed in canine and feline clinical samples should be selected as the EIPC to ensure the diagnostic reliability of the assay [16,17].

Since the emergence of SARS-CoV-2, the World Health Organization (WHO) has classified different isolates according to their pathogenic potential and virulence as variants of concern (VOC), variants of interest (VOI), and variants under monitoring (VUM) [18,19]. The sensitivity and specificity of RT-qPCR assays are mainly determined by the design of the primers and probes that specifically hybridize with conserved regions of the target viral gene sequences. The emergence of various SARS-CoV-2 genetic mutants can have a negative effect on the performance of existing RT-qPCR assays because genetic mutations in the primer/probe-binding sites of the target genes can result in potential mismatches, which can reduce diagnostic sensitivity and result in false-negative results [20,21,22]. Furthermore, a bioinformatic analysis study using 28 previously reported RT-qPCR assays and 793 publicly available SARS-CoV-2 genomes isolated from animals suggested the existence of at least one mismatch in the primer/probe binding sites of 16 of the 28 assays studied [23]. Therefore, considering the accumulated mutations in the SARS-CoV-2 genome and the emergence of various genetic variants, primers and probes for the newly developed RT-qPCR assay for companion animals should be carefully designed to match the target gene sequences of the currently prevalent SARS-CoV-2 variants in human and animal populations.

Most of the widely used RT-qPCR assays for human SARS-CoV-2 infections have been developed using primer and probe sets based on the *N*, *E*, and *RdRp* gene sequences. However, other related betacoronaviruses, such as bat SARS-like coronavirus and SARS-CoV genomes, have high sequence similarity to the *E*-gene assay primers and probe sequences [24]. Therefore, the initial screening with the *E*-gene assay, followed by a confirmation test with either the *RdRp* or the *N*-gene assays, are recommended [24,25,26,27]. The aim of this study was to develop a tailored multiplex RT-qPCR (mRT-qPCR) assay with species-specific EIPC for accurate and reliable diagnosis of SARS-CoV-2 infections in suspected companion animals (dogs and cats). To achieve this goal, we selected primer and probe sets that perfectly matched the conserved *RdRp* and *N* gene regions of currently prevalent SARS-CoV-2 isolates from human and animal hosts. Furthermore, we adopted a housekeeping gene stably expressed in canine and feline clinical samples as the EIPC instead of the human housekeeping gene in existing RT-qPCR assays for humans to improve the reliability of the developed RT-qPCR assay. Finally, the diagnostic performance of the developed RT-qPCR assay was comparatively evaluated with a commercial mRT-qPCR kit approved by emergency-use-authorization in Korea using nucleic acids extracted from different SARS-CoV-2 variants as well as canine and feline clinical samples. The developed mRT-qPCR assay with a canine or feline EIPC that simultaneously amplifies the *RdRp* and *N* genes of SARS-CoV-2 could serve as a promising diagnostic tool for SARS-CoV-2 detection in suspected canine and feline clinical cases.

## 2. Materials and Methods

### 2.1. Design of Primers and Probes for mRT-qPCR

SARS-CoV-2 genomic sequences, including VOC and VOI in human populations and animal isolates from different animal hosts, were obtained from the Global Initiative on Sharing All Influenza Data (GISAID) database, which covers all continents, including Europe, America, Asia, Africa, and Oceania (GISAID, 2020–2022; World Organisation for Animal Health, 2022; WHO, 2022). Sequences with high identity were excluded using the CD-HIT program (http://usegalaxy.eu/ accessed on 20 November 2022) with a threshold of 0.99, and 3084 sequences were ultimately used to design the primers and probes in this study (Appendix A).

The BioEdit sequence alignment editor program (http://www.mbio.ncsu.edu/BioEdit/bioedit.html accessed on 21 November 2022) was used to align the sequences, and conserved regions suitable for designing primer and probe sets were identified for the target *RdRp* and *N* gene sequences. Based on the conserved sequences, two sets of primers and probes were designed using Geneious Prime (Biomatters Ltd., Auckland, New Zealand) to specifically amplify the *RdRp* and *N* genes of SARV-CoV-2 (Appendix A). To check for the specificity of the designed primer and probe sets for SARS-CoV-2, the target sequences of the primers and probe sets were aligned with *RdRp* and *N* gene sequences of various SARS-CoV-2 variants. Furthermore, a BLAST search of the National Center for Biotechnology Information (NCBI) GenBank database (http://www.ncbi.nlm.nih.gov/BLAST/ accessed on 23 November 2022) was performed against random nucleotide sequences to confirm potential cross-reactivity of primers and probes for the SARS-CoV-2 *RdRp* and *N* genes. The OligoAnalyzer™ tool from Integrated DNA Technologies (Integrated DNA Technologies Inc., Iowa, USA) (https://sg.idtdna.com/pages/tools/oligoanalyzer accessed on 23 November 2022) was used to check for possible secondary structures between primers and probes, such as hairpins, self-dimers, and hetero-dimers. The SARS-CoV-2 primers and probes were designed, paying special attention to the selection of genomic regions that differed from other SARS-CoV relatives (GenBank accession Nos. AY613947.1, AY559094.1, AY502924.1, AY278491.2, NC_004718.3, AY502927.1, NC_019843, NC_002645, NC_005831, NC_006577, and NC_006213) and canine and feline coronaviruses (GenBank accession Nos. KP849472, KC175340, EU856362, JX860640, EU186072, DQ010921, DQ848678, and NC_002306) (Appendix A). These pre-evaluation processes helped design primers and probes specific to SARS-CoV-2 and avoided potential cross-reactions with other SARS-CoV relatives and canine and feline coronaviruses.

Furthermore, to avoid false-negative results, the canine or feline housekeeping gene *16S rRNA* was used as an EIPC marker for the presence of canine or feline cellular materials. For designing *16S rRNA*-specific primers and probes, 17 canine and 15 feline *16S rRNA* sequences were obtained from the NCBI GenBank database. Multiple alignments were performed using the BioEdit sequence alignment editor program to identify conserved nucleotide sequences within *16S rRNA* genes. Based on these conserved sequences, a pair of primers and probes was designed using the Geneious Prime software to detect *16S rRNA*. A BLAST search was used to confirm the specificity of the primers and probes for canine and feline *16S rRNA* using random nucleotide sequences.

For simultaneous and differential detection of SARS-CoV-2 *RdRp* and *N* genes, as well as canine or feline EIPC in a reaction, reporter dyes that are distinct or minimally overlap with the fluorescence spectra must be used to label the sequence-specific probes [28]. In this study, probes were labeled with different fluorescent dyes at both the 5′ and 3′ ends for simultaneous and differential detection of SARS-CoV-2 *RdRp* and *N* genes and canine or feline EIPC in a single reaction: cyanine 5 (Cy5) and Black Hole Quencher 2 (BHQ2) for the gene, 6-carboxyfluorescein (FAM) and Black Hole Quencher 1 (BHQ1) for the *N* gene, and 6-carboxy-2′,4,4′,5′,7,7′-hexachlorofluorescein (HEX) and BHQ1 for each EIPC, according to the manufacturer’s guidelines (BIONICS, Daejeon, Republic of Korea). The sequence, length, melting temperature (Tm), size, and genomic position of each primer and probe are listed in Table 1.

### 2.2. Samples and Nucleic Acid Extraction

Twenty-one SARS-CoV-2 strains, including VOCs (Alpha, Beta, Gamma, Delta, and Omicron) and VOIs (Iota, Lambda, Eta, Theta, Zeta, Epsilon, Kappa, and Mu variants) were obtained from the National Culture Collection for Pathogens (NCCP, Osong-eup, Chungcheongbuk-do, Republic of Korea) and used to develop and optimize the mRT-qPCR assay (Appendix A). A synthetic SARS-CoV plasmid (Tor2 strain; NCBI accession No. NC_004718), seven canine pathogens, including canine coronavirus (CCoV, NL-18 strain), canine distemper virus (CDV, Onderstepoort strain), canine adenovirus 2 (CAdV-2, Ditchfield strain), canine parvovirus (CPV, 7809 16-LP strain), canine influenza virus (CIV, A/Canine/Korea/01/07(H3N2) strain), canine parainfluenza virus (CPIV, D008 strain), and *Bordetella bronchiseptica* (*B. bronchiseptica*, S-55 strain), and three feline pathogens, including feline calicivirus (FCV, 894-T strain), feline herpesvirus (FHV, 593-J strain), and feline leukemia virus (FeLV, Rickard strain) were obtained from commercially available vaccines and used for the evaluation of the assay’s specificity (Table 2). For clinical evaluation of the mRT-qPCR assay, a total of 557 nasopharyngeal samples (280 dogs and 277 cats), including 37 SARS-CoV-2 positive samples (14 dogs and 23 cats), were obtained through collaboration with a companion animal healthcare company (Postbio Co., Ltd., Guri, Gyeonggi-do, Republic of Korea), a regional veterinary service laboratory (RVSL, Daegu, Republic of Korea), and the Animal and Plant Quarantine Agency (APQA, Kimcheon, Republic of Korea). Viral RNA was extracted from each sample (200 μL) using a TANBead nucleic acid extraction kit with an automated magnetic bead-based platform (Taiwan Advanced Nanotech Inc., Taoyuan, Taiwan) into elution buffer (100 μL), according to the manufacturer’s instructions. The extracted nucleic acids were allocated and stored at −80 °C until further use.

### 2.3. Reference Gene Construction for mRT-qPCR Analysis

The partial *RdRp* and *N* genes of SARS-CoV-2 spanning the amplified region of the developed mRT-qPCR assay were amplified by RT-PCR from RNA samples of the SARS-CoV-2 omicron variant (hCoV-19/Republic of Korea/KDCA18126/2021) using *RdRp* gene-specific primers (forward, 5′-GCCTCTTATTGTAACAGCTTTAAG-3′ and reverse, 5′-CAATTTGGGTGGTATGTCTGATC-3′) and *N* gene-specific primers (forward, 5′-ATGTCTGATAATGGACCCCAA-3′ and reverse, 5′-ACTTCCATGCCAATGCG-3′), which were designed based on the sequence of the omicron variant (GISAID accession ID: EPI_ISL_6959993). Reverse transcription and cDNA synthesis were performed using a commercial kit (PrimeScript™ first strand cDNA Synthesis Kit; Takara Bio Inc., Shiga, Japan). PCR was performed continually using a commercial kit (TaKaRa Ex Taq^®^; Takara Bio Inc., Shiga, Japan) in 50 μL reaction mixtures containing 5 μL of 10× Ex Taq Buffer, 4 μL dNTP mixture, 0.25 μL TaKaRa Ex Taq, 0.2 μM of each primer, and 5 μL of SARS-CoV-2 cDNA as a template, according to the manufacturer’s instructions. cDNA was amplified in a thermal cycler (Applied Biosystems, Foster City, CA, USA) under the following conditions: initial denaturation at 98 ℃ for 1 min, followed by 35 cycles of thermocycling (98 °C for 10 s, 58 °C for 30 s, and 72 °C for 2 min), and a final extension at 72 °C for 5 min. The amplified 1697 bp *RdRp* and 971 bp *N* gene sequences were inserted into the pTOP TA V2 vector (Enzynomics, Daejeon, Republic of Korea). The recombinant plasmid DNA samples were linearized using EcoRI and purified using the Expin CleanUP SV kit (GeneAll Biotechnology, Seoul, Republic of Korea). Subsequently, in vitro RNA transcription was performed using the RiboMAX Express Large Scale RNA Production System-T7 (Promega, Fitchburg, WI, USA) according to the manufacturer’s instructions. Using a NanoDrop Lite spectrophotometer (Thermo Fisher Scientific, Waltham, MA, USA), RNA concentration was determined by measuring the absorbance at 260 nm. After determining the RNA concentration, the copy numbers of the RNA transcript were quantified as previously described [29]. The RNA transcripts were serially diluted 10-fold (10^7^ to 1 copies/μL), stored at −80 °C, and used as an RNA standards for SARS-CoV-2 *RdRp* and *N* genes.

### 2.4. Optimization of mRT-PCR Conditions

Before optimizing the mRT-qPCR, a monoplex RT-qPCR assay was performed with each SARS-CoV-2 *RdRp* gene, *N* gene, or EIPC primer and probe set using a commercial RT-qPCR kit (RealHelix™ qRT-PCR Kit [v4], NanoHelix, Daejeon, Republic of Korea) and the CFX96 Touch™ Real-Time PCR Detection System (Bio-Rad, Hercules, CA, USA). The 25 μL reaction mixture containing 12.5 μL of 2× reaction buffer, 1 μL of 25× enzyme mix, 0.4 μM of each primer, 0.2 μM probe, 5 μL SARS-CoV-2 RNA template (10^6^–10^0^ copies/reaction), and SARS-CoV-2-negative canine and feline RNAs as the EIPC was prepared according to the manufacturer’s instructions. To optimize the mRT-qPCR conditions, the concentrations of the three sets of primers and probes were optimized, whereas the other reaction components were maintained identical to those used in monoplex RT-qPCR. The monoplex and multiplex RT-qPCR programs were the same and comprised 30 min at 50 °C for reverse transcription, 15 min at 95 °C for initial denaturation, followed by 40 cycles of 95 °C for 15 s and 60 °C for 60 s for amplification. Cy5 (SARS-CoV-2 *RdRp* gene), FAM (SARS-CoV-2 *N* gene), and HEX (EIPC) fluorescence signals for the tested samples were measured at the end of each annealing step. To interpret the monoplex RT-qPCR and mRT-qPCR results, samples with both *RdRp* and *N* gene cycle threshold (Ct) ≤ 37 were regarded as positive, whereas those with a higher Ct value (>37) were regarded as negative. The samples were considered invalid if EIPC was not detected within 40 amplification cycles.

### 2.5. Sensitivity of the mRT-PCR Assay

The analytical sensitivity of the mRT-qPCR assay for SARS-CoV-2 was determined using serial dilutions (10^6^–10^0^ copies/reaction) of each RNA standard of SARS-CoV-2 *RdRp* and *N* genes in triplicate. For data analysis, a standard curve of Ct values from 10-fold dilutions of SARS-CoV-2 *RdRp* and *N*-gene RNA standards (10^6^–10^0^ copies/reaction) was created using CFX96 Touch™ Real-Time PCR Detection software (Bio-Rad, Hercules, CA, USA). The correlation coefficient (*R*^2^) of the standard curve, the standard deviations of the results, and the SARS-CoV-2 RNA copy numbers in the samples were calculated from the standard curves using the detection software. The efficiency of the assay was determined using a previously described calculation [30].

### 2.6. Specificity of the mRT-qPCR Assay

To test the specificity of the mRT-qPCR assay, the assay was performed using RNA samples obtained from a SARS-CoV-2 omicron variant (GISAID accession ID. EPI_ISL_6959993), SARS-CoV (Tor2 strain), seven canine pathogens (CCoV, CDV, CAdV-2, CPV, CIV, CPIV, and *B.bronchiseptica*), three feline pathogens (FCV, FHV, and FeLV), two SARS-CoV-2-negative canine and feline clinical samples, two canine- or feline-origin cell cultures (MDCK and CRFK cells), and two non-canine- or feline-origin cell cultures (Vero and ST cells) as negative controls.

### 2.7. Precision of the mRT-qPCR Assay

The repeatability (intra-assay precision) and reproducibility (inter-assay precision) of the mRT-qPCR assay for SARS-CoV-2 detection were determined using three different concentrations (high, medium, and low) of each viral standard gene tested. The concentrations of SARS-CoV-2 *RdRp* and *N* genes were 10^6^, 10^4^, and 10^2^ copies/reaction. Using each dilution, the intra-assay variability was analyzed in triplicate on the same day, whereas the inter-assay variability was analyzed in six independent experiments performed by two experimenters on different days, per the MIQE (Minimum Information for Publication of Quantitative Real-Time PCR Experiments) guidelines [30]. The coefficient of variation (CV) for the Ct values was determined from the results of the intra- or inter-assay experiments and expressed as a percentage of the mean value, accompanied by the standard deviation values.

### 2.8. Reference RT-qPCR Assays

A commercially available mRT-qPCR kit (PowerChek™ SARS-CoV-2 Real-time PCR Kit, Kogenebiotech, Seoul, Republic of Korea) was used to compare the diagnostic performance of the mRT-qPCR assay developed in this study. Commercial mRT-qPCR (Kogene) was performed with SARS-CoV-2 *RdRp* and *E* genes and human *GAPDH* gene-specific primer and probe sets using the CFX96 Touch™ Real-time PCR Detection System (Bio-Rad, Hercules, CA, USA) according to the manufacturer’s instructions. Real-time fluorescence values of the FAM- (*RdRp* gene), HEX- (*E* gene), and Cy5 (IC)-labeled probes were measured in ongoing reactions at the end of each annealing step. To interpret the mRT-qPCR results, both *RdRp* and *E* genes with Ct ≤ 38 were considered positive. The sample was considered negative if either no Ct values were observed after the completion of 40 cycles of amplification or if Ct values were >38. Retesting was recommended if the *RdRp* or *E* gene was a single positive.

### 2.9. Diagnostic Performance of the mRT-qPCR Assay 

The diagnostic performance of the mRT-qPCR assay was evaluated in three steps with different categories of samples. First, to evaluate the diagnostic sensitivity of the mRT-qPCR assay for different SARS-CoV-2 variants circulating in the human population, RNA samples of 21 SARS-CoV-2 variants isolated from human clinical cases in the Korean epidemic were obtained from NCCP (Appendix A) and tested using the newly developed and Kogene’s mRT-qPCR assays using the same concentration of the viral RNA templates (10^4^ RNA copies/reaction), and the results were compared. Second, to evaluate the diagnostic performance of the mRT-qPCR assay for SARS-CoV-2-positive animal samples, 37 RNA samples extracted from SARS-CoV-2-positive canine and feline clinical samples (14 dogs and 23 cats) were obtained from APQA and tested using the developed mRT-qPCR and Kogene’s mRT-qPCR assays, and the results were compared. Finally, to evaluate the diagnostic performance of mRT-qPCR for blind clinical samples, a total of 520 nasopharyngeal samples (266 dogs and 254 cats) were obtained from a companion animal health-care company (Postbio Co., Ltd., Guri, Gyeonggi-do, Republic of Korea) and tested using the developed mRT-qPCR and Kogene’s mRT-qPCR assays, and the results were compared. Based on the results of both the assays for canine and feline clinical samples (37 positive samples and 520 blind samples), the inter-assay concordance was analyzed using Cohen’s kappa statistic with a 95% confidence interval (CI). The interpretation of the calculated kappa coefficient value (κ) was as follows: κ < 0.20 = slight agreement, 0.21–0.40 = fair agreement, 0.41–0.60 = moderate agreement, 0.61–0.80 = substantial agreement, and 0.81–1.0 = almost perfect agreement [31].

## 3. Results

### 3.1. Interpretation of the mRT-qPCR Assay

The fluorescent signals of FAM for the SARS-CoV-2 *N* gene, Cy5 for the SARS-CoV-2 *RdRp* gene, and HEX for EIPC were successfully generated by mRT-qPCR with each primer and probe set and the corresponding SARS-CoV-2 RNA or canine and feline RNA samples (Figure 1). The results of mRT-qPCR using the optimized primer and probe concentration (0.4 µM of each primer and 0.2 µM of each probe for SARS-CoV-2 *RdRp, N,* and EIPC) suggested that three fluorescent signals of FAM, Cy5, and HEX could be simultaneously detected in a single reaction. Furthermore, HEX signals for EIPC were consistently detected for nasopharyngeal samples, regardless of the SARS-CoV-2 standard RNA concentration (Figure 1E). These results suggested that mRT-qPCR could successfully amplify the two target genes of SARS-CoV-2 and canine or feline *16S rRNA* in clinical samples in a single reaction without significant crosstalk or spurious amplification among the three fluorescent reporter dyes. The efficiency of the mRT-PCR assay for SARS-CoV-2 *RdRp* and *N* genes was 100.7% and 96.3%, respectively, which were comparable to those of the corresponding monoplex RT-qPCR for *RdRp* (98.9%) and *N* (100.7%) genes (Figure 1).

### 3.2. Specificity of the mRT-qPCR Assay

The established SARS-CoV-2 *RdRp* and *N* gene-specific primer and probe sets generated positive Cy5 and FAM signals with a VOC strain only. No positive signals were generated with other canine and feline pathogens or cell cultures. Using the mRT-qPCR with primers and probes for canine or feline EIPC, positive HEX signals were detected for canine or feline clinical samples, canine- or feline-origin cells, and canine or feline live attenuated vaccines. In addition, no SARS-CoV-2 or EIPC positive signals (Cy5, FAM, and HEX) were detected in the two non-canine- or feline-origin cells (Table 2). These results indicate that the established mRT-qPCR assay was highly specific to SARS-CoV-2 RNAs and can be considered reliable, as it did not provide false-negative results.

### 3.3. Sensitivity of the mRT-PCR Assay

The analytical sensitivity of the mRT-qPCR assay was determined using 10-fold serial dilutions of the SARS-CoV-2 standard RNAs. The limit of detection (LOD) of mRT-qPCR was determined to be below 10 copies/reaction for the SARS-CoV-2 *RdRp* and *N* genes, which were similar to those of the monoplex RT-qPCR (Figure 1A,C,E). Standard curves for targeted genes were generated by plotting their Ct values against their dilution factors, to determine PCR efficiency and the linearity of the reaction. High correlation values (*R*^2^ > 0.99) were found between the Ct values and dilution factors for the monoplex RT-qPCR and mRT-qPCR assays (Figure 1B,D,F). 

### 3.4. Precision of the mRT-qPCR Assay

To assess the mRT-qPCR precision, three different concentrations (high, medium, and low) for each standard RNA were tested in triplicate in six different runs performed by two experimenters on different days. The coefficients of variation within runs (intra-assay variability) ranged from 0.25% to 0.58% for *RdRp* and 0.41% to 0.54% for the *N* gene. In contrast, inter-assay variability ranged from 0.51% to 0.66% for *RdRp* and 0.22% to 0.59% for the *N* gene, respectively (Table 3). These results demonstrate that the developed mRT-qPCR method is accurate and reliable for detecting SARS-CoV-2.

### 3.5. Comparative Clinical Evaluation of the mRT-qPCR Assay

To evaluate the diagnostic sensitivity of the developed mRT-qPCR assay, we first tested 21 RNA samples (all adjusted to 10^4^ copies/reaction) extracted from SARS-CoV-2 variants, and the results were compared with those of the commercially available mRT-qPCR assay using *RdRp* and *E* gene-specific primers and probe sets. For identical RNA concentration of all tested viruses, the mean Ct values by the developed mRT-qPCR assay were determined to be 33.09 (32.14–35.36) for the *RdRp* gene and 32.06 (31.02–34.20) for the *N* gene; using Kogene’s mRT-qPCR, the mean Ct values were 33.31 (32.03–35.15) for the *RdRp* gene and 34.19 (33.16–36.24) for the *E* gene (Appendix A). The Ct values produced by the developed mRT-qPCR assay for *RdRp* and *N* genes were slightly lower than those produced by Kogene’s mRT-qPCR, which indicates that the developed mRT-qPCR assay is more sensitive than Kogene’s mRT-qPCR assay for the detection of various SARS-CoV-2 variants.

Subsequently, 557 canine and feline clinical samples (280 dogs and 277 cats), including 37 known SARS-CoV-2 positive samples (14 dogs and 23 cats), were tested using the developed mRT-qPCR and Kogene’s RT-qPCR assays. In both assays, 37 known SARS-CoV-2-positive samples were determined to be SARS-CoV-2 RNA-positive, and the remaining 520 clinical samples were determined to be SARS-CoV-2-negative (Table 4). Based on the clinical evaluation results, the detection rates of both assays were the same, at 6.6% (5.0% in dogs and 8.3% in cats). The percentage of positive, negative, and overall agreement between the results of the developed and Kogene’s mRT-qPCRs were 100% (37/37), 100% (520/520), and 100% (557/557), respectively. The kappa value (95% CI) between mRT-qPCR and Kogene’s mRT-qPCR was 1.0, indicating that the diagnostic results of the developed and Kogene’s mRT-qPCR assays were 100% concordant and exhibited complete agreement (Table 4). Furthermore, using the established mRT-qPCR assay, EIPC (canine or feline *16S rRNA*) signals were generated in all tested clinical samples, except in four clinical samples (one dog and three cats), indicating that canine or feline cellular material was not included or degraded in the four clinical samples. The samples were unsuitable for molecular diagnosis. These results suggest that the developed mRT-qPCR assay is valuable for the clinical diagnosis of canine and feline SARS-CoV-2 infection.

## 4. Discussion

Given that the viral agent of COVID-19, SARS-CoV-2, is a zoonotic as well as reverse zoonotic agent, cooperative efforts of medical and veterinary scientists are required to control this ongoing pandemic. One of the biggest concerns in the field of veterinary medicine in response to COVID-19 is the emergence of new variants through host adaptation to infected animals such as companion dogs and cats, having close contact with their owners, which may pose a threat to animal and public health [10,11,12]. The first step in combating the emerging virus is extensive surveillance and rapid detection of infected companion animals, as well as appropriate isolation and treatment. To achieve this goal, secure diagnostic methods are required for rapid, accurate, and reliable detection of the virus in suspected companion animals. However, to the best of our knowledge, no mRT-qPCR method tailored to detecting SARS-CoV-2 infections in canine and feline clinical cases has been described. Therefore, in the present study, we developed an mRT-qPCR assay using three sets of primers and probes to hybridize the conserved *N* and *RdRp* genes of SARS-CoV-2 and canine or feline *16S rRNA* gene as an EIPC for the reliable diagnosis of SARS-CoV-2 infection in suspected canine and feline clinical cases.

This newly developed mRT-qPCR assay has several advantages. Currently, RT-qPCR is a reliable diagnostic method widely used for detecting symptomatic and asymptomatic patients infected with SARS-CoV-2 owing to high sensitivity and specificity. However, the sensitivity and specificity of the RT-qPCR assay may be negatively affected by potential preanalytical and analytical vulnerabilities, including improper sample collection and processing as well as lack of harmonization of primers and probes with target genes [15,32,33]. Potential mismatches and false negatives may occur owing to the continued emergence of SARS-CoV-2 mutants and the resulting genetic variation in the viral genome at the primer/probe binding sites of an RT-qPCR assay. Most widely used RTq-PCR assays have mismatches between sequences of their primers and probes and currently prevailing SARS-CoV-2 strains in human and animal populations [20,21,22,23]. Therefore, a more efficient mRT-qPCR method based on primers and probes carefully designed to detect SARS-CoV-2 strains currently circulating in human and animal populations should be developed for diagnosing SARS-CoV-2 infection in companion dogs and cats. In this regard, the main source of transmission of SARS-CoV-2 to companion animals was the virus-infected owners living with companion animals; however, the possibility of animal-to-animal transmission cannot be excluded under special circumstances, such as in animal shelters [7,8,9,34]. In this study, to ensure high diagnostic sensitivity for detecting currently circulating SARS-CoV-2 variants, we carefully designed primer and probe sets based on the highly conserved *RdRp* and *N* gene sequences of 3084 SARS-CoV-2 strains, including various human-origin variants and animal-origin isolates (Appendix A). The sequences of the designed primers and probes completely matched the target gene sequences of SARS-CoV-2 strains. They had relatively low nucleotide identity with SARS-CoV, Middle East respiratory syndrome-related coronavirus, and other human and animal coronaviruses (Appendix A). The developed mRT-qPCR with the designed primers and probes specifically amplified SARS-CoV-2 RNAs but did not amplify nucleic acids from other canine and feline pathogens, indicating that the newly designed primer/probe sets were highly specific to SARS-CoV-2 (Table 2). Moreover, a comparative evaluation of the assay with 21 Korean SARS-CoV-2 isolates indicated that the assay successfully detected all tested viruses with relatively low Ct values compared to those of Kogene’s mRT-qPCR assay (Appendix A), indicating that the developed mRT-qPCR assay has a high diagnostic sensitivity for detecting prevalent Korean SARS-CoV-2 strains.

The LODs of the developed mRT-qPCR assay were below 10 copies/reaction for SARS-CoV-2 *RdRp* and *N* genes (Figure 1). Considering that the LODs of WHO-listed SARS-CoV-2 RT-qPCR assays range from 100 to 2.7 copies/reaction [14] and the LODs of three commercial SARS-CoV-2 RT-qPCR kits approved by emergency-use authorization in Korea, Allplex (Seegene, Seoul, Republic of Korea), PowerChek (KogeneBiotech, Seoul, Republic of Korea), and Real-Q (BioSewoom, Seoul, Republic of Korea), were determined to be 153.9, 84.1, and 80.6 copies/reaction, respectively [35], the sensitivity of the developed mRT-qPCR assay is suitable to diagnose SARS-CoV-2 in companion animals. In addition to high sensitivity, coefficients of variation of intra-assay repeatability and inter-assay reproducibility of the developed mRT-qPCR assay were <1.0% (Table 3), demonstrating that the mRT-qPCR assay developed in this study is an accurate and reliable diagnostic tool for SARS-CoV-2.

Most currently available mRT-qPCR methods to detect human SARS-CoV-2 infection commonly use human housekeeping genes as EIPC to discriminate false negative results during clinical diagnosis [15,19,36]. Using these methods to diagnose SARS-CoV-2 in suspected dogs and cats, as is currently being performed by animal disease diagnostic laboratories, makes it impossible to monitor false-negative results using the assay, which can raise critical problems in the control of viral infections. Therefore, including canine or feline EIPC in the developed mRT-qPCR assay for SARS-CoV-2 detection from canine and feline clinical samples is required to monitor the reliability and accuracy of the assay results. To improve the reliability of the mRT-qPCR assay developed in this study, we used the canine and feline housekeeping gene *16S rRNA* as an EIPC, which is present in most types of canine and feline samples, such as blood, urine, nasal swabs, and tissue biopsy, and no additional steps are required to prepare the internal control preparation or spike inoculation [17,37]. In this study, we newly designed a pair of primers that can simultaneously amplify canine and feline *16S rRNA* gene fragments and two species-specific probes that can detect each amplified canine or feline *16S rRNA* gene (Table 1), which allowed each EIPC to be amplified with the same efficiency in canine or feline clinical samples. The performance of canine and feline *16S rRNA* (EIPC) in mRT-qPCR was evaluated analytically, indicating that EIPC does not interact with the SARS-CoV-2 targets or affect the sensitivity or amplification efficiency of the assay for SARS-CoV-2 detection (Figure 1). Canine and feline *16S rRNA* (EIPC) were amplified using mRT-qPCR in all tested canine and feline clinical samples except four; thus, invalid samples could be filtered out to ensure the high reliability of the developed mRT-qPCR assay (Table 4). In the clinical evaluation of canine and feline clinical samples, the clinical diagnostic sensitivity of the developed mRT-qPCR assay was 6.6% (37/557) and was in 100% agreement with that of the commercial Kogene mRT-qPCR kit (Table 4). However, canine and feline EIPC was successfully amplified by the developed mRT-qPCR in most canine and feline clinical samples but not by Kogene’s mRT-qPCR. These results demonstrate that the developed mRT-qPCR assay is highly recommended as a molecular diagnostic tool for diagnosing SARS-CoV-2 infection in suspected canine and feline clinical cases. To the best of our knowledge, this is the first study to develop a tailored mRT-qPCR with high sensitivity, specificity, and reliability for diagnosing SARS-CoV-2 infection in companion dogs and cats. However, our study has a limitation: considering the genetic diversity and continuous evolution of SARS-CoV-2 during the ongoing pandemic, the possibility of further mutation in the gene sequence of the primer and probe binding sites cannot be ignored. Therefore, an appropriate diagnostic assay must be selected for currently circulating viral strains in human and animal populations and improving the primers and probes of the mRT-qPCR assay with continuous monitoring of viral mutations for accurate and sensitive diagnosis of SARS-CoV-2 infection.

## 5. Conclusions

In conclusion, the developed mRT-qPCR assay with high sensitivity, specificity, and reliability may be a promising molecular diagnostic tool for detecting SARS-CoV-2 in companion dogs and cats and will be useful for etiological diagnosis, epidemiological studies, and controlling SARS-CoV-2 infection in canine and feline populations.

## Figures and Tables

**Figure 1 animals-13-00602-f001:**
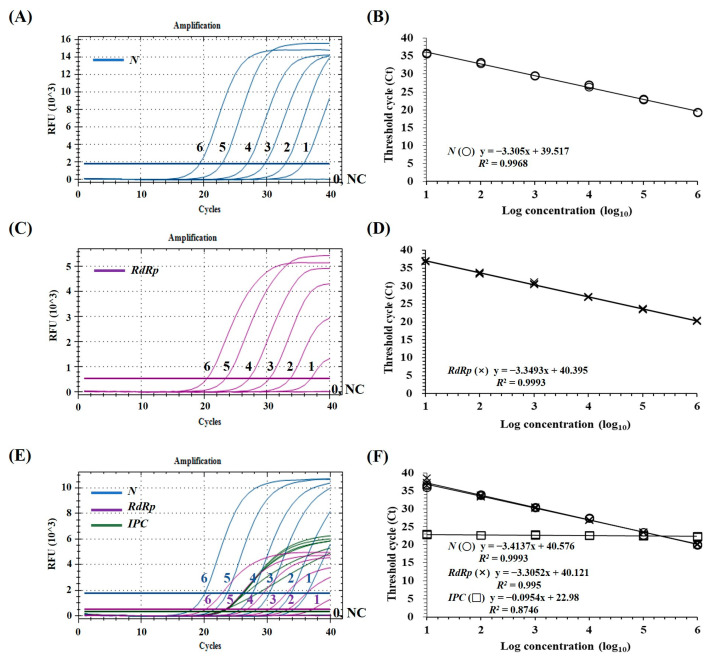
The limit of detection (LOD) and standard curve of the monoplex real-time RT-qPCR and multiplex real-time mRT-qPCR. The LOD and standard curve of the *N* and *RdRp* genes were obtained by monoplex RT-qPCR for 10-fold serial dilutions of SARS-CoV-2 standard RNA (**A**–**D**). The LOD and standard curves of mRT-qPCR for 10-fold serial dilutions of SARS-CoV-2 standard RNA are shown (**E**,**F**). Lines 6–0, 10-fold serial dilutions of the SARS-CoV-2 standard RNA (10^6^–10^0^ copies); NC, negative control. The coefficient of determination (*R^2^*) and the equation of the regression curve (y) were calculated using the CFX Manager software (Bio-Rad).

**Table 1 animals-13-00602-t001:** Primers and probes used in the developed mRT-qPCR.

Primer/Probe	Sequence (5′–3′)	Length (bp)	Tm (°C)	Genome Position ^a^
*RdRp*-F	GGTAACTGGTATGATTTCGGT	21	59.9	14080–14100
*RdRp*-R	CTGGTCAAGGTTAATATAGGCA	22	59.9	14165–14186
*RdRp*-P	Cy5-CACGCCAGGTAGTGGAGTTCCTGTT-BHQ2	25	69.2	14115–14139
*N*-F	CAGGAACTAATCAGACAAGGA	21	59.3	29138–29158
*N*-R	CGACATTCCGAAGAACGC	18	60.6	29210–29227
*N*-P	FAM-ATTGCACAATTTGCCCCCAGCG-BHQ1	22	68.5	29183–29204
*IPC*-F	AGACGAGAAGACCCTATG	18	57.1	2142–2159
*IPC*-R	GGTCACCCCAACCTAAAT	18	58.8	2240–2257
*IPC*-P (canine)	HEX-ACCTACAAGGCATAACATAACACCA-BHQ1	25	64.4	2199–2223
*IPC*-P (feline)	HEX-AAGAGACCATATGAACCAACCGACA-BHQ1	25	65.8	2848–2872

^a^ Genome position of primer and probe sequences according to SARS-CoV-2, canine *16S rRNA*, and feline *16S rRNA* sequences (GenBank accession Nos. NC_045512, MZ042325, and NC_028310, respectively). FAM, 6-carboxyfluorescein; BHQ1, Black Hole Quencher 1; Cy5, cyanine 5; BHQ2, Black Hole Quencher 2; HEX, 6-carboxy-2′,4,4′,5′,7,7′-hexachlorofluorescein; Tm, melting temperature.

**Table 2 animals-13-00602-t002:** Specificity of the mRT-qPCR assay using SARS-CoV-2 and EIPC-specific primer and probe sets.

Pathogen (Strain)	Source ^a^	Amplification of Target Gene ^b^			
		*RdRp* (Cy5)	*N* (FAM)	Canine EIPC (HEX)	Feline EIPC (HEX)
SARS-CoV-2 (hCoV-19/Republic of Korea/KDCA18126/2021)	NCCP	+	+	−	−
SARS-CoV (Tor2)	ADIC	−	−	−	−
Canine coronavirus (NL-18)	CAVS	−	−	−	−
Canine distemper virus (Onderstepoort)	CAVS	−	−	+	−
Canine adenovirus 2 (Ditchfield)	CAVS	−	−	+	−
Canine parvovirus (7809 16-LP)	CAVS	−	−	+	−
Canine influenza virus [A/Canine/Korea/01/07(H3N2)]	CAVS	−	−	+	−
Canine parainfluenza virus (D008)	CAVS	−	−	+	−
*Bordetella bronchiseptica* (S-55)	CAVS	−	−	+	−
Feline calicivirus (894-T)	CAVS	−	−	−	+
Feline herpesvirus 1 (593-J)	CAVS	−	−	−	+
Feline leukemia virus (Rickard)	CAVS	−	−	−	+
Non-infected canine swab sample	ADIC	−	−	+	−
Non-infected feline swab sample	ADIC	−	−	−	+
MDCK cells	ADIC	−	−	+	−
CRFK cells	ADIC	−	−	−	+
Vero cells	ADIC	−	−	−	−
ST cells	ADIC	−	−	−	−

^a^ NCCP, National Culture Collection for Pathogens (Republic of Korea); ADIC, Animal Disease Intervention Center (Kyungpook National University, Republic of Korea); CAVS, commercially available vaccine strain (Republic of Korea); EIPC, endogenous internal positive control; +: positive reaction; −: negative reaction. ^b^ HEX fluorescence signals for canine or feline EIPC were obtained from canine or feline pathogens, canine or feline clinical samples, and cells of canine/feline origin.

**Table 3 animals-13-00602-t003:** Intra- and inter-assay coefficient of variation of mRT-qPCR.

Target Gene	Dilution (Copies/Reaction)	Intra-Assay			Inter-Assay		
		Mean	SD	CV (%)	Mean	SD	CV (%)
*RdRp*	High (10^6^)	20.20	0.11	0.53	20.20	0.10	0.51
	Medium (10^4^)	26.87	0.07	0.25	26.82	0.14	0.53
	Low (10^2^)	33.50	0.19	0.58	33.98	0.23	0.66
*N*	High (10^6^)	20.03	0.11	0.54	20.22	0.08	0.42
	Medium (10^4^)	27.48	0.12	0.43	28.15	0.06	0.22
	Low (10^2^)	33.82	0.14	0.41	35.07	0.21	0.59

The mean value (mean), standard deviation (SD), and coefficient of variation (CV) were determined based on the cycle threshold (Ct) values by mRT-qPCR using three different concentrations of standard RNAs of SARS-CoV-2 RNA-dependent RNA polymerase (*RdRp*) and nucleocapsid (*N*) genes.

**Table 4 animals-13-00602-t004:** Comparison of the results of the developed and Kogene’s mRT-qPCR assays for detecting SARS-CoV-2 in canine and feline clinical samples.

New mRT-qPCR		Kogene’s mRT-qPCR			Detection Rate	Overall Percent Agreement
		Positive	Negative ^a^	Total		
Canine Samples	Positive	14	0	14	5.0%	100.0%
	Negative	0	266	266		
	Total	14	266	280		
Feline samples	Positive	23	0	23	8.3%	100.0%
	Negative	0	254	254		
	Total	23	254	277		
Total	Positive	37	0	37	6.6%	100.0%
	Negative	0	520	520		
	Total	37	520	557		

^a^ HEX signals for EIPCs (canine or feline *16S rRNA*) were not generated in one canine and three feline clinical samples.

## Data Availability

Not applicable.

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
