# Peer review of "Tailored Multiplex Real-Time RT-PCR with Species-Specific Internal Positive Controls for Detecting SARS-CoV-2 in Canine and Feline Clinical Samples"

_animals, 2023, doi:10.3390/ani13040602_

Round 1

Reviewer 1 Report

Paper is well written and I enjoyed reading it! Thank you.

Author Response

Dear reviewer

We appreciate reviewers for taking the time and effort to review our manuscript. We have addressed the reviewer’s comments. Below are our point-by-point responses to the reviewer’s comments.

Point 1: Paper is well written and I enjoyed reading it! Thank you.

Response 1: Thank you for the valuable comments. We’re glad that reviewer enjoyed our paper.

Thank you very much for kind comments of reviewers to our paper.

Reviewer 2 Report

Review Report

ID Animals - 2150390

Title: „Tailored multiplex real-time RT-PCR with species-specific internal positive controls for detecting SARS-CoV-2 in canine and feline clinical samples

Authors: Gyu-Tae Jeon, Hye-Ryung Kim, Jong-Min Kim, Ji-Su Baek, Yeun-Kyung Shin, Oh-Kyu Kwon, Hae-Eun Kang, Ho-Seong Cho, Doo-Sung Cheon, and Choi-Kyu Park

Version: 1/ Date: 04-01-2023

Reviewer number: 1

A brief summary

The manuscript „Tailored multiplex real-time RT-PCR with species-specific internal positive controls for detecting SARS-CoV-2 in canine and feline clinical samplesby Gyu-Tae Jeon, Hye-Ryung Kim, Jong-Min Kim, Ji-Su Baek, Yeun-Kyung Shin, Oh-Kyu Kwon, Hae-Eun Kang, Ho-Seong Cho, Doo-Sung Cheon, and Choi-Kyu Park describes the relatively new diagnostic possibilities of SARS-CoV-2 in canine and feline clinical samples, using an adapted multiplex real-time RT-PCR assay method with newly designed primers and probes targeting RdRp and N genes of SARS-CoV-2. The research data emphasizes the comparative analysis of the reaction results with particular attention to high sensitivity, specificity, accuracy, and reliability of the developed assay and its application in the field of laboratory diagnostics of COVID-19.

The objective (one main sentence) of the Article is not clearly identified at the end of the “Introduction” part. The other structural parts of the article Methods/ Materials, Results and Discussion are described clearly and in a sufficiently informative way. The Results are presented in a clear and consistent manner, the tables and figures are informative, some some statistical (basic) elements are presented in the text of the Results. The results of the research are relatively compared with the data of other scientists in the Discussion section, some elements of comparative analyses and personal opinion can be found.

Broad comments

The article's idea, descriptive elements, and potential added value for the laboratory diagnosis of SARS-CoV-2 in canine and feline clinical samples is particularly valuable, as the lack of such data raises many controversial debates, especially about the zoonotic potential of COVID-19 and the virus-risk factors for interspecies transmission of viruses in companion animal populations.

The Abstract is well prepared – no objective comments.

The Introduction prepared well (in advance) with possibly focus on the research topics of study, but the objective (one main sentence) is not clearly identified at the end of this part (remember the strategic definitions of the goal and tasks).

The Materials and Methods described in detail, although the criteria for the choice of methods could be detailed according to the work tasks. Statistical analysis additional information should be provided in this section – the selection criteria for groups comparison and the calculation of the correlation (between and within).

The Results are well presented and sufficiently understandable.

All the presented research results should be discussed in the article (Reference gene construction, Sensitivity, Precision). If it is not possible to compare such results with other authors data, the personal interpretations are strongly recommended. 

The list of reference should be revised and unified according to the specific recommendation for authors.

Specific comments

There are some specific recommendations based on my personal opinion:

L 47-123. The Introduction part should be shortened and be optimized: could start from L55 or L63; L84-91 could be reviewed or removed (or explain how it relates to the objective). The objective must be clearly identified.

L 161.  Ref. or http. prot. after „program“.

Additionally. All the first-time abbreviation must be explained (Reffer) in the text (…is important for a ,,unspecialized,, readers). The list of abbreviations should be prepared and added.

Author Response

Dear reviewer

We appreciate the reviewer for the time and effort dedicated to review our manuscript. We have addressed the reviewer’s comments and revised the manuscript accordingly. Below are our point-by-point responses to the reviewer’s comments.

Reviewer 3 Report

I liked this paper very much. Kudos to authors for presenting such a nice and important to the field work!

Author Response

Dear reviewer

We appreciate reviewers for taking the time and effort to review our manuscript. We have addressed the reviewer’s comments. Below are our point-by-point responses to the reviewer’s comments.

Point 1: I liked this paper very much. Kudos to authors for presenting such a nice and important to the field work!

Response 1: Thank you for the valuable comments. We’re glad that reviewer enjoyed our paper.

Thank you very much for kind comments of reviewers to our paper.